# Geometric Algebra Transformers

Johann Brehmer    Pim de Haan    Sönke Behrends    Taco Cohen

Qualcomm AI Research[1]

{jbrehmer, pim, sbehrend, tacos}@qti.qualcomm.com

*Abstract*—**Problems involving geometric data arise in a variety of fields, including computer vision, robotics, chemistry, and physics. Such data can take numerous forms, such as points, direction vectors, planes, or transformations, but to date there is no single architecture that can be applied to such a wide variety of geometric types while respecting their symmetries. In this paper we introduce the Geometric Algebra Transformer (GATr), a general-purpose architecture for geometric data. GATr represents inputs, outputs, and hidden states in the projective geometric algebra, which offers an efficient 16-dimensional vector space representation of common geometric objects as well as operators acting on them. GATr is equivariant with respect to $E(3)$, the symmetry group of 3D Euclidean space. As a transformer, GATr is scalable, expressive, and versatile. In experiments with $n$-body modeling and robotic planning, GATr shows strong improvements over non-geometric baselines.**

## I. Introduction

From molecular dynamics to astrophysics, from material design to robotics, fields across science and engineering deal with geometric data: points, directions, surfaces, orientations, and so on. The geometric nature of data provides a rich structure: a notion of common operations between geometric types (computing distances between points, applying rotations to orientations, etc.), a well-defined behaviour of data under transformations of a system, and the independence of certain properties of coordinate system choices.

When learning relations from geometric data, incorporating this rich structure into the architecture has the potential to improve the performance, especially in the low-data regime. To implement such an inductive bias, it is useful to first categorize inputs, outputs, and internal data into certain object types, for instance group representations. Next, the functions mapping between these types have certain regularity constraints imposed, for instance based on equivariance [6].

In this spirit, we introduce the *Geometric Algebra Transformer* (GATr), a general-purpose network architecture for geometric data. GATr brings together three key ideas.

**Geometric algebra:** To naturally describe both geometric objects as well as their transformations in three-dimensional space, GATr represents data as multivectors of the projective geometric algebra $\mathbb{G}_{3,0,1}$. Geometric algebra is an elegant, versatile and practical mathematical framework for geometrical computations. The particular algebra $\mathbb{G}_{3,0,1}$ extends the vector space $\mathbb{R}^3$ to 16-dimensional multivectors, which can natively represent various geometric types and $E(3)$ poses. In this framework, common interactions between

geometric data types can be computed with few operations, in particular the geometric product.

**Equivariance:** To behave consistently under transformations, GATr is equivariant with respect to $E(3)$, the symmetry group of three-dimensional space. To this end, we develop several new $E(3)$-equivariant primitives mapping between multivectors, including equivariant linear maps, an attention mechanism, nonlinearities, and normalization layers.

**Transformer:** Due to its favorable scaling properties, expressiveness, trainability, and versatility, the transformer architecture [23] has become the de-facto standard for a wide range of problems. GATr is based on the transformer architecture, and hence inherits these benefits.

GATr hence combines two lines of research: the representation of geometric objects with geometric algebra [9, 10, 18], popular in computer graphics and physics and recently gaining traction in deep learning [3, 19, 21], and the encoding of symmetries through equivariant deep learning [7]. The result—to the best of our knowledge the first $E(3)$-equivariant architecture with internal geometric algebra representations—is a versatile network for problems involving geometric data. We demonstrate GATr in a robotic planning problem, where it significantly outperforms non-geometric baselines.

## II. Geometric algebra in a nutshell

We begin with the briefest of introductions to geometric algebra. For an in-depth introduction, we point the interested reader to Refs. [9, 10, 18, 19].

Whereas a plain vector space like $\mathbb{R}^3$ allows us to take linear combinations of elements $x$ and $y$ (vectors), a geometric algebra additionally has a bilinear associative operation: the *geometric product*, denoted simply by $xy$. By multiplying vectors, one obtains so-called *multivectors*, which can represent both geometrical *objects* and *operators*. Multivectors can be expanded on a multivector basis, characterized by their dimensionality or grade, such as scalars (grade 0), vectors $e_i$ (grade 1), bivectors $e_i e_j$ (grade 2), all the way up to the *pseudoscalar* $e_1 \cdots e_d$ (grade $d$). The symmetric and antisymmetric parts of the geometric product are called the interior and exterior (wedge) product. Finally, we will require is the dualization operator $x \mapsto x^*$. It acts on basis elements by swapping "empty" and "full" dimensions, e.g. sending $e_1 \mapsto e_{23}$.

In order to represent three-dimensional objects as well as arbitrary rotations and translations acting on them, we work with the projective geometric algebra $\mathbb{G}_{3,0,1}$ [9, 18, 19]. Here one adds a fourth *homogeneous coordinate* $x_0 e_0$ to the 3D

---

[1]Qualcomm AI Research is an initiative of Qualcomm Technologies, Inc.

| Object / operator | Scalar | Vector | | Bivector | | Trivector | | PS |
| --- | --- | --- | --- | --- | --- | --- | --- | --- |
| | $1$ | $e_0$ | $e_i$ | $e_{0i}$ | $e_{ij}$ | $e_{0ij}$ | $e_{123}$ | $e_{0123}$ |
| Scalar $\lambda \in \mathbb{R}$ | $\lambda$ | $0$ | $0$ | $0$ | $0$ | $0$ | $0$ | $0$ |
| Plane w/ normal $n \in \mathbb{R}^3$, origin shift $d \in \mathbb{R}$ | $0$ | $d$ | $n$ | $0$ | $0$ | $0$ | $0$ | $0$ |
| Line w/ direction $n \in \mathbb{R}^3$, orthogonal shift $s \in \mathbb{R}^3$ | $0$ | $0$ | $0$ | $s$ | $n$ | $0$ | $0$ | $0$ |
| Point $p \in \mathbb{R}^3$ | $0$ | $0$ | $0$ | $0$ | $0$ | $p$ | $1$ | $0$ |
| Pseudoscalar $\mu \in \mathbb{R}$ | $0$ | $0$ | $0$ | $0$ | $0$ | $0$ | $0$ | $\mu$ |
| Reflection through plane w/ normal $n \in \mathbb{R}^3$, origin shift $d \in \mathbb{R}$ | $0$ | $d$ | $n$ | $0$ | $0$ | $0$ | $0$ | $0$ |
| Translation $t \in \mathbb{R}^3$ | $1$ | $0$ | $0$ | $\frac{1}{2}t$ | $0$ | $0$ | $0$ | $0$ |
| Rotation expressed as quaternion $q \in \mathbb{R}^4$ | $q_0$ | $0$ | $0$ | $0$ | $q_i$ | $0$ | $0$ | $0$ |
| Point reflection through $p \in \mathbb{R}^3$ | $0$ | $0$ | $0$ | $0$ | $0$ | $p$ | $1$ | $0$ |

TABLE I: Embeddings of common geometric objects and transformations into the projective geometric algebra $\mathbb{G}_{3,0,1}$. The columns show different components of the multivectors with the corresponding basis elements, with $i, j \in \{1, 2, 3\}, j \neq i$, i.e. $ij \in \{12, 13, 23\}$. For simplicity, we fix gauge ambiguities (the weight of the multivectors) and leave out signs (which depend on the ordering of indices in the basis elements).

vector space, yielding a $2^4 = 16$-dimensional geometric algebra. The metric of $\mathbb{G}_{3,0,1}$ is such that $e_0^2 = 0$ and $e_i^2 = 1$ for $i = 1, 2, 3$.

We can use $\mathbb{G}_{3,0,1}$ to represent transformations: a vector $u$ represents the reflection of other elements in the hyperplane orthogonal to $u$. Since any orthogonal transformation is equal to a sequence of reflections, this allows us to express any such transformation as a geometric product of (unit) vectors, $u = u_1 \cdots u_k$. These form the *Pin group*, which turns out to be the double cover of E(3). In order to apply elements of the Pin group to an arbitrary multivector $x$, one uses the *sandwich product*:

$$\rho_u(x) = \begin{cases} uxu^{-1} & \text{if } u \text{ is even} \\ u\hat{x}u^{-1} & \text{if } u \text{ is odd} \end{cases} \quad (1)$$

Here $\hat{x}$ is the *grade involution*, which flips the sign of odd-grade elements such as vectors and trivectors, while leaving even-grade elements unchanged.

Following Refs. [9, 18, 19], we represent planes with vectors, and require that the intersection of two geometric objects is given by the wedge product of their representations. Lines (the intersection of two planes) are thus represented as bivectors, points (the intersection of three planes) as trivectors. This leads to a duality between objects and operators, where objects are represented like transformations that leave them invariant. Table I provides a dictionary of these embeddings. It is easy to check that this representation is consistent with using the sandwich product for transformations.

We construct network layers that are equivariant with respect to E(3), or equivalently its double cover Pin(3, 0, 1). A function $f : \mathbb{G}_{3,0,1} \to \mathbb{G}_{3,0,1}$ is Pin(3, 0, 1)-equivariant with respect to the representation $\rho$ (or Pin(3, 0, 1)-equivariant for short) if $f(\rho_u(x)) = \rho_u(f(x))$ for any $u \in \text{Pin}(3, 0, 1)$ and $x \in \mathbb{G}_{3,0,1}$.

## III. THE GEOMETRIC ALGEBRA TRANSFORMER

### A. Architecture overview

The Geometric Algebra Transformer (GATr) is designed based on three principles outlines in the introduction: a strong inductive bias for geometric data through a representation based on geometric algebra, symmetry awareness through E(3) equivariance, and scalability and versatility through a transformer architecture.

We sketch GATr in Fig. 1. In the top row, we show the overall workflow. If necessary, raw inputs are first preprocessed into geometric types. The geometric objects are then embedded into multivectors of the geometric algebra $\mathbb{G}_{3,0,1}$, following the recipe described in Tbl. I.

The multivector-valued data are processed with a GATr network. We show this architecture in more detail in the bottom row of Fig. 1. GATr consists of $N$ transformer blocks, each consisting of an equivariant multivector LayerNorm, an equivariant multivector self-attention mechanism, a residual connection, another equivariant LayerNorm, an equivariant multivector MLP with geometric bilinear interactions, and another residual connection. The architecture is thus similar to a typical transformer [23] with pre-layer normalization [1, 24], but adapted to correctly handle multivector data and be E(3) equivariant. We describe the individual layers below.

Finally, from the outputs of the GATr network we extract the target variables, again following the mapping given in Tbl. I.

### B. GATr primitives

*a) Linear layers:* We begin with linear layers between multivectors. In Appendix A, we show that the equivariance condition severely constrains them:

**Proposition 1.** *Any linear map $\phi : \mathbb{G}_{d,0,1} \to \mathbb{G}_{d,0,1}$ that is equivariant to $\text{Pin}(d, 0, 1)$ is of the form*

$$\phi(x) = \sum_{k=0}^{d+1} w_k \langle x \rangle_k + \sum_{k=0}^{d} v_k e_0 \langle x \rangle_k \quad (2)$$

*for parameters $w \in \mathbb{R}^{d+2}, v \in \mathbb{R}^{d+1}$. Here $\langle x \rangle_k$ is the blade projection of a multivector, which sets all non-grade-$k$ elements to zero.*

Thus, E(3)-equivariant linear maps between $\mathbb{G}_{3,0,1}$ multivectors can be parameterized with nine coefficients, five of which are the grade projections and four include a multiplication with the homogeneous basis vector $e_0$. We thus parameterize affine layers between multivector-valued arrays with Eq. (2), with learnable coefficients $w_k$ and $v_k$ for each combination of input

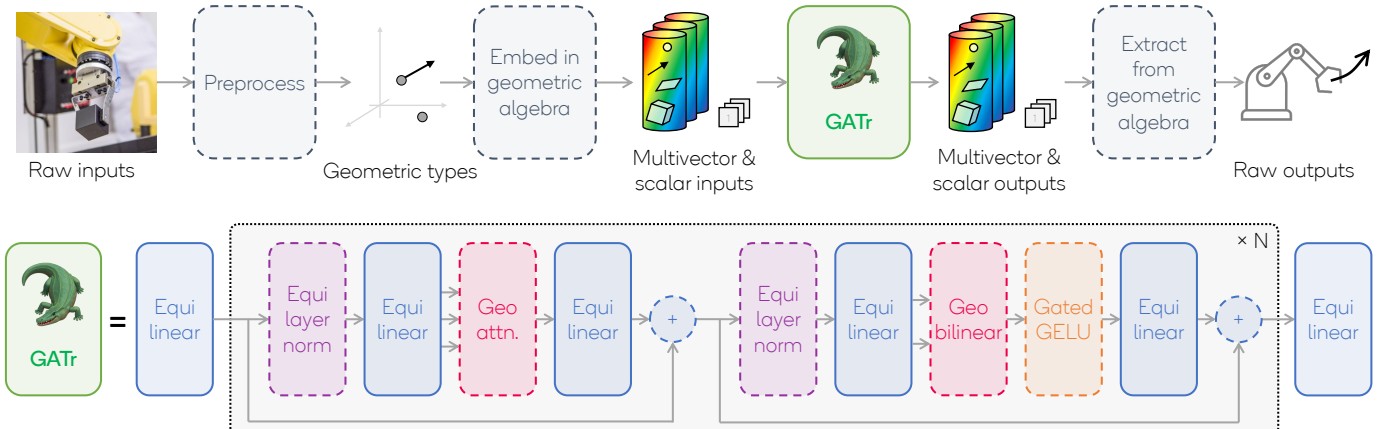

Fig. 1: Overview over the GATr architecture. Boxes with solid lines are learnable components, those with dashed lines are fixed.

channel and output channel. In addition, there is a learnable bias term for the scalar components of the outputs (biases for the other components are not equivariant).

*b) Geometric bilinears:* Equivariant linear maps are not sufficient to build expressive networks. The reason is that these operations allow for only very limited grade mixing. For the network to be able to construct new geometric features from existing ones, such as the translation vector between two points, two additional primitives are essential.

The first is the geometric product $x, y \mapsto xy$, the fundamental bilinear operation of geometric algebra. It allows for substantial mixing between grades: for instance, the geometric product of vectors consists of scalars and bivector components. The geometric product is equivariant (Appendix A).

The second geometric primitive we use derived from the so-called *join*[2] $x, y \mapsto (x^* \wedge y^*)^*$. This map may appear complicated, but it plays a simple role in our architecture: an equivariant map that involves the dual $x \mapsto x^*$. Including the dual in an architecture is essential for expressivity: in $\mathbb{G}_{3,0,1}$, without any dualization it is impossible to represent even simple functions such as the Euclidean distance between two points [9]; we show this in Appendix A. While the dual itself is not $\mathrm{Pin}(3, 0, 1)$-equivariant (w. r. t. $\rho$), the join operation is equivariant to even (non-mirror) transformations. To make the join equivariant to mirrorings as well, we multiply its output with a pseudoscalar derived from the network inputs: $x, y, z \mapsto \mathrm{EquiJoin}(x, y, z) = z_{0123}(x^* \wedge y^*)^*$, where $z_{0123} \in \mathbb{R}$ is the pseudoscalar component of a reference multivector $z$.

We define a *geometric bilinear layer* that combines the geometric product and the join of the two inputs as $\mathrm{Geometric}(x, y; z) = \mathrm{Concatenate}_{\mathrm{channels}}(xy, \mathrm{EquiJoin}(x, y; z))$. In GATr, this layer is included in the MLP.

*c) Nonlinearities and normalization:* We use scalar-gated GELU nonlinearities [12] $\mathrm{GatedGELU}(x) = \mathrm{GELU}(x_1)x$, where $x_1$ is the scalar component of the multivector $x$. Moreover, we define an E(3)-equivariant LayerNorm operation

[2]Technically, the join has an anti-dual, not the dual, in the output. We leave this detail out for notational simplicity.

for multivectors as $\mathrm{LayerNorm}(x) = x/\sqrt{\mathbb{E}_c \langle x, x \rangle}$, where the expectation goes over channels and we use the invariant inner product $\langle \cdot, \cdot \rangle$ of $\mathbb{G}_{3,0,1}$.

*d) Attention:* Given multivector-valued query, key, and value tensors, each consisting of $n_i$ items (or tokens) and $n_c$ channels (key length), we define the E(3)-equivariant multivector attention as

$$\mathrm{Attention}(q, k, v)_{i'c'} = \sum_i \mathrm{Softmax}_i \left( \frac{\sum_c \langle q_{i'c}, k_{ic} \rangle}{\sqrt{8n_c}} \right) v_{ic'}. \tag{3}$$

Here the indices $i, i'$ label items, $c, c'$ label channels, and $\langle \cdot, \cdot \rangle$ is the invariant inner product of the geometric algebra. Just as in the original transformer [23], we thus compute scalar attention weights with a scaled dot product; the difference is that we use the inner product of $\mathbb{G}_{3,0,1}$. We extend this attention mechanism to multi-head self-attention in the usual way.

## C. Extensions

*a) Auxiliary scalar representations:* While multivectors are well-suited to model geometric data, many problems contain non-geometric information as well. Such scalar information may be high-dimensional, for instance in sinosoidal positional encoding schemes. Rather than embedding into the scalar components of the multivectors, we add an auxiliary scalar representation to the hidden states of GATr. Each layer thus has both scalar and multivector inputs and outputs. They have the same batch dimension and item dimension, but may have different number of channels.

This additional scalar information interacts with the multi-vector data in two ways. In linear layers, we allow the auxiliary scalars to mix with the scalar component of the multivectors. In the attention layer, we compute attention weights both from the multivectors, as given in Eq. (3), and from the auxiliary scalars, using a regular scaled dot-product attention. The two attention maps are summed before computing the softmax, and the normalizing factor is adapted. In all other layers, the scalar information is processed separately from the multivector information, using the unrestricted form of the multivector map. For instance, nonlinearities transform multivectors with

| Method | Reward |
|---|---|
| GATr-Diffuser (ours) | **74.8** ± 1.7 |
| Transformer-Diffuser | 69.8 ± 1.9 |
| Diffuser [15] (reproduced) | 57.7 ± 1.8 |
| Diffuser [15] | 58.7 ± 2.5 |
| EDGI [5] | 62.0 ± 2.1 |
| CQL [17] | 24.4 |
| BCQ [11] | 0.0 |

TABLE II: Diffusion-based robotic planning. We show the normalized cumulative rewards achieved on a robotic block stacking task [15], where 100 is optimal and means that each block stacking task is completed successfully, while 0 corresponds to a failure to stack any blocks. We show the mean and standard error over at least 100 evaluation episodes. The top three results were computed in the GATr code base, the bottom four taken from the literature [5, 15].

equivariant gated GELUs and auxiliary scalars with regular GELU functions.

*b) Rotary positional embeddings:* GATr assumes the data can be described as a set of items (or tokens). If these items are distinguishable and form a sequence, we encode their position using rotary position embeddings [22] in the auxiliary scalar variables.

*c) Axial attention over objects and time:* The architecture is flexible about the structure of the data. In some use cases, there will be a single dimension along which objects are organized, for instance when describing a static scene or the time evolution of a single object. But GATr also supports the organization of a problem along multiple axes, for example with one dimension describing objects and another time steps. In this case, we follow an axial transformer layout [13], alternating between transformer blocks that attend over different dimensions. (The not-attended dimensions in each block are treated like a batch dimension.)

## IV. ROBOTIC PLANNING THROUGH INVARIANT DIFFUSION

In Appendix C, we demonstrate Kuka on a synthetic $n$-body regression problem. We find that it outperforms non-geometric baselines and the $E(3)$-equivariant SEGNN in terms of sample efficiency and generalization.

In this section of the main paper, we restrict ourselves to a robotics experiment. We show how GATr defines an $E(3)$-invariant diffusion model, that it can be used for model-based reinforcement learning and planning, and that this combination is well-suited to solve robotics problems.

We follow Janner et al. [15], who propose to treat learning a world model and planning within that model as a unified generative modeling problem. After training a diffusion model [20] on offline trajectories, one can use it in a planning loop, sampling from it conditional on the current state, desired future states, or to maximize a given reward, as needed.

We embed a GATr model in this algorithm and call this combination *GATr-Diffuser*. GATr is equivariant with respect to $E(3)$ and the object permutation group $S_n$. When used together with a base density that is $E(3) \times S_n$-invariant, the diffusion model is also $E(3) \times S_n$-invariant [2, 16]. Often, a particular task requires breaking this symmetry: imagine, for instance, that a particular object needs to be moved to

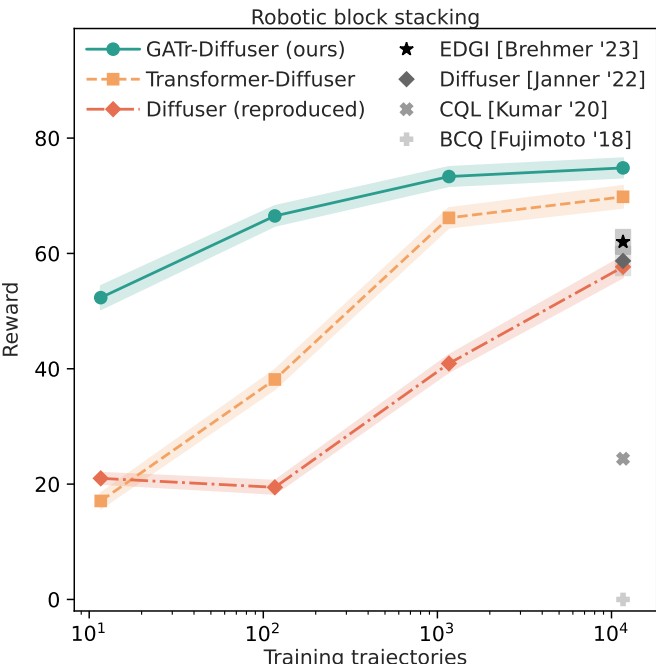

Fig. 2: Diffusion-based robotic planning. We show normalized rewards (higher is better) as in Tbl. II as a function of training dataset size. GATr (———) is more successful at block stacking and more sample-efficient than the baselines, including the original Diffuser model [15] (—·—) and our modification of it based on a Transformer (— —). In grey, we show results reported in the literature [5, 15].

a particular location. The Diffuser approach is an excellent match for such situations, as conditioning on the current state, future state, or a reward model as proposed by Janner et al. [15] can softly break the symmetry group as desired [5].

GATr-Diffuser is demonstrated on the problem of a Kuka robotic gripper stacking blocks using the "unconditional" environment introduced by Janner et al. [15]. We train a GATr-Diffuser model on the offline trajectory dataset published with that paper. To facilitate a geometric interpretation, we parameterize the data in terms of geometric quantities like object positions and orientations. In particular, we use the position and pose of the robotic endeffector as features and map to joint angles with an inverse kinematics model. We then test GATr-Diffuser on its ability to stack four blocks on each other. We compare our GATr-Diffuser model to a reproduction of the original Diffuser model (based on the published code, but using our data parameterization) and a new transformer backbone for the Diffuser model. In addition, we show the published results of Diffuser [15], the equivariant EDGI [5], and the offline RL algorithms CQL [17] and BCQ [11] as published in Ref. [15]. The problem and hyperparameters are described in detail in Appendix D.

As shown in Tbl. II and Fig. 2, GATr-Diffuser is able to solve the block-stacking problem better than all baselines. It is also clearly more sample-efficient, matching the performance of a Diffuser model trained on the full dataset even when training only on 1% of the trajectories. The fact that GATr-Diffuser also outperforms the $E(3)$-equivariant EDGI model [5] is evidence that equivariance alone is not the key to its success, hinting that the geometric algebra provides a useful inductive bias.

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

## A. Theoretical results

In this section, we state or prove several properties of equivariant maps between geometric algebras that we use in the construction of GATr.

The grade involution is a linear involutive bijection $\widehat{\cdot}$ : $\mathbb{G}_{n,0,r} : \mathbb{G}_{n,0,r}$, which sends a $k$-blade $x$ to $\widehat{x} = (-1)^k x$. Note that this is an algebra automorphism $\widehat{xy} = \widehat{x}\widehat{y}$, and also an $\wedge$-algebra automorphism. The reversal in a linear involutive bijection $\widetilde{\cdot} : \mathbb{G}_{n,0,r} : \mathbb{G}_{n,0,r}$ which sends a $k$-blade $x = x_1 \wedge x_2 \wedge ... \wedge x_k$ to the reverse: $\widetilde{x} = x_k \wedge ... \wedge x_2 \wedge x_1 = \pm x$ with $+x$ if $k \in \{0, 1, 4, 5, ..., 8, 9, ...\}$ and $-x$ otherwise. Note that this is an anti-automorphism (contravariant functor): $\widetilde{xy} = \widetilde{y}\widetilde{x}$.

Here we denote the sandwich action of $u \in \text{Pin}(n, 0, r)$ on a multivector $x$ not as $\rho_u(x)$, but as $u[x]$. For odd $u$, $u[x] = u\widehat{x}u^{-1}$, while for even $u$, $u[x] = uxu^{-1}$. The sandwich action is linear by linearity of the $\widehat{\cdot}$ and bilinearity of the geometric product. Furthermore, note that for any particular $u \in \text{Pin}(n, 0, r)$, the action is a geometric algebra homomorphism: $u[ab] = uabu^{-1} = u\widehat{a}u^{-1}u\widehat{b}u^{-1} = u[a]u[b]$. By linearity and a symmetrization argument [10, Sec 7.1], one can show that it also a $\wedge$-algebra homomorphism (outermorphism): $u[a \wedge b] = u[a] \wedge u[b]$.

Let $l \geq k$. Given a $k$-vector $a$ and $l$-vector $b$, define the *left contraction* as $a \rfloor b := \langle ab \rangle_{l-k}$, which is a $l - k$-vector. For $k = 1$, and $b$ a blade $b = b_1 \wedge ... \wedge b_l$. Geometrically, $a \rfloor b$ is the projection of $a$ to the space spanned by the vectors $b_i$. Thus we have that $a \rfloor b = 0 \iff \forall i, \langle a, b_i \rangle = 0$ [10, Sec 3.2.3], in which case we define $a$ and $b$ to be *orthogonal*. In particular, two vectors $a, b$ are orthogonal if their inner product is zero. Futhermore, we define a vector $a$ to be *tangential* to blade $b$ if $a \wedge b = 0$.

In the projective algebra, a blade $x$ is defined to be *ideal* if it can be written as $x = e_0 \wedge y$ for another blade $y$.

*1) Linear maps:* We begin with Pin-equivariant linear maps. After some technical lemmata, we prove the most general form of linear equivariant maps in the Euclidean geometric algebra $\mathbb{G}_{n,0,0}$, and then also in projective geometric algebra $\mathbb{G}_{n,0,1}$.

**Proposition 2.** *The grade projection $\langle \cdot \rangle_k$ is equivariant [10, Sec 13.2.3].*

*Proof:*

Choose an $l$-blade $x = a_1 \wedge a_2 \wedge ... \wedge a_l$. Let $u$ be a 1-versor. As the action $u$ is an outermorphism, $u[x] = u[a_1] \wedge ... \wedge u[a_l]$ is an $l$-blade. Now if $l \neq k$, then $\langle x \rangle_k = 0$ and thus $u[\langle x \rangle_k] = \langle u[x] \rangle_k$. If $l = k$, then $\langle x \rangle_k = x$ and thus $u[\langle x \rangle_k] = \langle u[x] \rangle_k$. As the grade projection is linear, equivariance extends to any multivector. ∎

**Proposition 3.** *The following map is equivariant: $\phi : \mathbb{G}_{3,0,1} \to \mathbb{G}_{3,0,1} : x \mapsto e_0 x$.*

*Proof: Let $u$ be a 1-versor, then $u$ acts on a multivector as $x \mapsto u[x] = u\hat{x}u^{-1}$, where $\hat{x}$ is the grade involution. Note that $e_0$ is invariant: $u[e_0] = -ue_0 u^{-1} = e_0 uu^{-1} = e_0$, where $ue_0 = -e_0 u$ because $u$ and $e_0$ are orthogonal: $ue_0 =*

$\langle u, e_0 \rangle + u \wedge e_0 = -e_0 \wedge u = -e_0 \wedge u$. *Then $\phi$ is equivariant, as the action is an algebra homomorphism: $u[\phi(x)] = u[e_0 x] = \widehat{ue_0 x}u^{-1} = u\hat{e}_0 u^{-1} u\hat{x}u^{-1} = u[e_0]u[x] = e_0 u[x] = \phi(u[x])$. It follows that $\phi$ is also equivariant to any product of vectors, i.e. any versor $u$.* ∎

*a) Euclidean geometric algebra:* Before constructing the most general equivariant linear map between multivectors in projective geometric algebra, we begin with the Euclidean case $\mathbb{G}_{n,0,0}$.

**Theorem 1** (Cartan-Dieuodonné). *Every orthogonal transformation of an $n$-dimensional space can be decomposed into at most $n$ reflections in hyperplanes.*

*Proof: This theorem is proven in Roelfs and De Keninck [18].* ∎

**Lemma 1.** *In the $n$-dimensional Euclidean geometric algebra $\mathbb{G}_{n,0,0}$, the group $\text{Pin}(n, 0, 0)$ acts transitively on the space of $k$-blades of norm $\lambda \in \mathbb{R}^{>0}$.*

*Proof: As the $\text{Pin}$ group preserves norm, choose $\lambda = 1$ without loss of generality. Any $k$-blade $x$ of unit norm can be written by Gram-Schmidt factorization as the wedge product of $k$ orthogonal vectors of unit norm $x = v_1 \wedge v_2 \wedge ... \wedge v_k$. Consider another $k$-blade $y = w_1 \wedge w_2 \wedge ... \wedge w_k$ with $w_i$ orthonormal. We'll construct a $u \in \text{Pin}(n, 0, 0)$ such that $u[x] = y$.*

*Choose $n - k$ additional orthonormal vectors $v_{k+1}, ..., v_n$ and $w_{k+1}, .., .w_n$ to form orthonormal bases. Then, there exists a unique orthogonal transformation $\mathbb{R}^n \to \mathbb{R}^n$ that maps $v_i$ into $w_i$ for all $i \in \{1, ..., n\}$. By the Cartan-Dieuodonné theorem 1, this orthogonal transformation can be expressed as the product of reflections, thus there exists a $u \in \text{Pin}(n, 0, 0)$ such that $u[v_i] = w_i$. As the $u$ action is a $\wedge$-algebra homomorphism ($u[a \wedge b] = u[a] \wedge u[b]$, for any multivectors $a, b$), we have that $u[x] = y$.* ∎

**Lemma 2.** *In the Euclidean ($r = 0$) or projective ($r = 1$) geometric algebra $\mathbb{G}_{n,0,r}$, let $x$ be a $k$-blade. Let $u$ be a 1-versor. Then $u[x] = x \iff u \rfloor x = 0$ and $u[x] = -x \iff u \wedge x = 0$.*

*Proof: Let $x$ be a $k$-blade and $u$ a vector of unit norm. We can decompose $u$ into $u = t + v$ with $t \wedge x = 0$ (the part tangential to the subspace of $x$) and $v \rfloor x = 0$ (the normal part). This decomposition is unique unless $x$ is ideal in the projective GA, in which case the $e_0$ component of $u$ is both normal and tangential, and we choose $t$ Euclidean.*

*In either case, note the following equalities: $xt = (-1)^{k-1}tx; xv = (-1)^k vx; vt = -tv$ and note $\nexists \lambda \neq 0$ such that $vtx = \lambda x$, which can be shown e.g. by picking a basis. Then:*

$$\begin{aligned} u[x] &= (-1)^k (t + v)x(t + v) \\ &= (t + v)(-t + v)x \\ &= (-\|t\|^2 + \|v\|^2)x - 2vtx. \end{aligned}$$

*We have $u[x] \propto x \iff vtx = 0$. If $x$ is not ideal, this implies that either $v = 0$ (thus $u \wedge x = 0$ and $u[x] = -x$) or $t = 0$*

*(thus $u \rfloor x = 0$ and $u[x] = x$). If $x$ is ideal, this implies that either $v \propto e_0$ (thus $u \wedge x = 0$ and $u[x] = -x$) or $t = 0$ (thus $u \rfloor x = 0$ and $u[x] = x$).* ∎

**Lemma 3.** *Let $r \in \{0, 1\}$. Any linear $\mathrm{Pin}(n, 0, r)$-equivariant map $\phi : \mathbb{G}_{n,0,r} \to \mathbb{G}_{n,0,r}$ can be decomposed into a sum of equivariant maps $\phi = \sum_{lkm} \phi_{lkm}$, with $\phi_{lkm}$ equivariantly mapping $k$-blades to $l$-blades. If $r = 0$ (Euclidean algebra) or $k < n + 1$, such a map $\phi_{lkm}$ is defined by the image of any one non-ideal $k$-blade, like $e_{12...k}$. Instead, if $r = 1$ (projective algebra) and $k = n + 1$, then such a map is defined by the image of a pseudoscalar, like $e_{01...n}$.*

*Proof: The $\mathrm{Pin}(n, 0, r)$ group action maps $k$-vectors to $k$-vectors. Therefore, $\phi$ can be decomposed into equivariant maps from grade $k$ to grade $l$: $\phi(x) = \sum_{lk} \phi_{lk}(\langle x \rangle_k)$, with $\phi_{lk}$ having $l$-vectors as image, and all $k'$-vectors in the kernel, for $k' \neq k$. Let $x$ be an non-ideal $k$-blade (or pseudoscalar if $k = n + 1$). By lemmas 1 and 4, in both Euclidean and projective GA, the span of the $k$-vectors in the orbit of $x$ contains any $k$-vector. So $\phi_{lk}$ is defined by the $l$-vector $y = \phi_{lk}(x)$. Any $l$-vector can be decomposed as a finite sum of $l$-blades: $y = y_1 + ... y_M$. We can define $\phi_{lkm}(x) = y_m$, extended to all $l$-vectors by equivariance, and note that $\phi_{lk} = \sum_m \phi_{lkm}$.* ∎

**Proposition 4.** *For an $n$-dimensional Euclidean geometric algebra $\mathbb{G}_{n,0,0}$, any linear endomorphism $\phi : \mathbb{G}_{n,0,0} \to \mathbb{G}_{n,0,0}$ that is equivariant to the $\mathrm{Pin}(n, 0, 0)$ group (equivalently to $O(n)$) is of the type $\phi(x) = \sum_{k=0}^{n} w_k \langle x \rangle_k$, for parameters $w \in \mathbb{R}^{n+1}$.*

*Proof: By decomposition of Lemma 3, let $\phi$ map from $k$-blades to $l$-blades. Let $x$ be a $k$-blade. Let $u$ be a 1-versor. By Lemma 2, if $u$ is orthogonal to $x$, $u[\phi(x)] = \phi(u[x]) = \phi(x)$ and $u$ is also orthogonal to $\phi(x)$. If $u \wedge x = 0$, then $u[\phi(x)] = \phi(u[x]) = \phi(-x) = -\phi(x)$ and $u \wedge \phi(x) = 0$. Thus any vector in $x$ is in $\phi(x)$ and any vector orthogonal to $x$ is orthogonal to $\phi(x)$, this implies $\phi(x) = w_k x$, for some $w_k \in \mathbb{R}$. By Lemma 3, we can extend $\phi$ to $\phi(y) = w_k y$ for any $k$-vector $y$.* ∎

*b) Projective geometric algebra:* How about equivariant linear maps in *projective* geometric algebra? The degenerate metric makes the derivation more involved, but in the end we will arrive at a result that is only slightly more general.

**Lemma 4.** *The Pin group of the projective geometric algebra, $\mathrm{Pin}(n, 0, 1)$, acts transitively on the space of $k$-blades with positive norm $\|x\| = \lambda > 0$. Additionally, the group acts transitively on the space of zero-norm $k$-blades of the form $x = e_0 \wedge y$ (called ideal blades), with $\|y\| = \kappa$.*

*Proof: Let $x = x_1 \wedge ... \wedge x_k$ be a $k$-blade with positive norm $\lambda$. All vectors $x_i$ can be written as $x_i = v_i + \delta_i e_0$, for a nonzero Euclidean vector $v_i$ (meaning with no $e_0$ component) and $\delta_i \in \mathbb{R}$, because if $v_i = 0$, the norm of $x$ would have been 0. Orthogonalize them as $x'_2 = x_2 - \langle x_1, x_2 \rangle x_1$, etc., resulting in $x = x'_1 \wedge \cdots \wedge x'_k$ with $x'_i = v'_i + \delta'_i e_0$ with orthogonal $v'_i$. Define the translation $t = 1 + \frac{1}{2} \sum_i \delta'_i e_0 \wedge v'_i$, which makes $x'$ Euclidean: $t[x'] = v'_1 \wedge ... \wedge v'_k$. By Lemma 1, the Euclidean*

*Pin group $\mathrm{Pin}(n, 0, 0)$, which is a subgroup of $\mathrm{Pin}(n, 0, 1)$, acts transitively on Euclidean $k$-blades of a given norm. Thus, in the projective geometric algebra $\mathrm{Pin}(n, 0, 1)$, any two $k$-blades of equal positive norm $\lambda$ are related by a translation to the origin and then a $\mathrm{Pin}(n, 0, 0)$ transformation.*

*For the ideal blades, let $x = e_0 \wedge y$, with $\|y\| = \kappa$. We take $y$ to be Euclidean without loss of generality. For any $g \in \mathrm{Pin}(n, 0, 1)$, $g[e_0] = e_0$, so $g[x] = e_0 \wedge g[y]$. Consider another $x' = e_0 \wedge y'$ with $\|y'\| = \kappa$ and taking $y'$ Euclidean. As $\mathrm{Pin}(n, 0, 0)$ acts transitively on Euclidean $(k - 1)$-blades with norm $\kappa$, let $g \in \mathrm{Pin}(n, 0, 0)$ such that $g[y] = y'$. Then $g[x] = x'$.* ∎

We can now construct the most general equivariant linear map between projective geometric algebras, a key ingredient for GATr:

**Proposition 5.** *For the projective geometric algebra $\mathbb{G}_{n,0,1}$, any linear endomorphism $\phi : \mathbb{G}_{n,0,1} \to \mathbb{G}_{n,0,1}$ that is equivariant to the group $\mathrm{Pin}(n, 0, r)$ (equivalently to $E(n)$) is of the type $\phi(x) = \sum_{k=0}^{n+1} w_k \langle x \rangle_k + \sum_{k=0}^{n} v_k e_0 \langle x \rangle_k$, for parameters $w \in \mathbb{R}^{n+2}, v \in \mathbb{R}^{n+1}$.*

*Proof: Following Lemma 3, decompose $\phi$ into a linear equivariant map from $k$-blades to $l$-blades. For $k < n + 1$, let $x = e_{12...k}$. Then following Lemma 2, for any $1 \le i \le k$, $e_i \wedge x = 0, e_i[x] = -x$, and $e_i[\phi(x)] = \phi(e_i[x]) = \phi(-x) = -\phi(x)$ and thus $e_i \wedge \phi(x) = 0$. Therefore, we can write $\phi(x) = x \wedge y_1 \wedge ... \wedge y_{l-k}$, for $l - k$ vectors $y_j$ orthogonal to $x$.*

*Also, again using Lemma 2, for $k < i \le n$, $e_i \rfloor x = 0 \implies e_i[\phi(x)] = \phi(x) \implies e_i \rfloor \phi(x) = 0 \implies \forall i, \langle e_i, y_j \rangle = 0$. Thus, $y_j$ is orthogonal to all $e_i$ with $1 \le i \le n$. Hence, $l = k$ or $l = k + 1$ and $y_1 \propto e_0$.*

*For $k = n + 1$, let $x = e_{012...k}$. By a similar argument, all invertible vectors $u$ tangent to $x$ must be tangent to $\phi(x)$, thus we find that $\phi(x) = x \wedge y$ for some blade $y$. For any non-zero $\phi(x)$, $y \propto 1$, and thus $\phi(x) \propto x$. By Lemma 3, by equivariance and linearity, this fully defines $\phi$.* ∎

*2) Bilinear maps:* Next, we turn towards bilinear operations. In particular, we show that the geometric product and the join are equivariant.

For the geometric product, equivariance is straightforward: Any transformation $u \in \mathrm{Pin}(n, 0, r)$, gives a homomorphism of the geometric algebra, as for any multivectors $x, y$, $u[xy] = u\widehat{xy}u^{-1} = u\widehat{x}\widehat{y}u^{-1} = u\widehat{x}u^{-1}u\widehat{y}u^{-1} = u[x]u[y]$. The geometric product is thus equivariant.

*a) Dual and join in Euclidean algebra:* For the join and the closely related dual, we again begin with the Euclidean geometric algebra, before turning to the projective case later.

The role of the dual is to have a bijection $\cdot^* : \mathbb{G}_{n,0,0} \to \mathbb{G}_{n,0,0}$ that maps $k$-vectors to $(n-k)$-vectors. For the Euclidean algebra, with a choice of pseudoscalar $\mathcal{I}$, we can define a dual as:

$$x^* = x\mathcal{I}^{-1} = x\tilde{\mathcal{I}} \tag{4}$$

This dual is bijective, and involutive up to a sign: $(y^*)^* = y\tilde{\mathcal{I}}\tilde{\mathcal{I}} = \pm y$, with $+y = 1$ for $n \in \{1, 4, 5, 8, 9, ...\}$ and $-y$ for $n \in \{2, 3, 6, 7, ...\}$. We choose $\tilde{\mathcal{I}}$ instead of $\mathcal{I}$ in the definition

of the dual so that given $n$ vectors $x_1, ..., x_n$, the dual of the multivector $x = x_1 \wedge ... x_n$, is given by the scalar of the oriented volume spanned by the vector. We denote the inverse of the dual as $x^{-*} = x\mathcal{I}$. Expressed in a basis, the dual yields the complementary indices and a sign. For example, for $n = 3$ and $\mathcal{I} = e_{123}$, we have $(e_1)^* = -e_{23}$, $(e_{12})^* = e_3$.

Via the dual, we can define the bilinear join operation, for multivectors $x, y$:

$$x \vee y := (x^* \wedge y^*)^{-*} = ((x\tilde{\mathcal{I}}) \wedge (y\tilde{\mathcal{I}}))\mathcal{I}.$$

**Lemma 5.** *In Euclidean algebra $\mathbb{G}_{n,0,0}$, the join is $\mathrm{Spin}(n, 0, 0)$ equivariant. Furthermore, it is $\mathrm{Pin}(n, 0, 0)$ equivariant if and only if $n$ is even.*

*Proof: The join is equivariant to the transformations from the group $\mathrm{Spin}(n, 0, 0)$, which consists of the product of an even amount of unit vectors, because such transformations leave the pseudoscalar $\mathcal{I}$ invariant, and the operation consists otherwise of equivariant geometric and wedge products.*

*However, let $e_{12...n} = \mathcal{I} \in \mathrm{Pin}(n, 0, 0)$ be the point reflection, which negates vectors of odd grades by the grade involution: $\mathcal{I}[x] = \hat{x}$. Let $x$ be a $k$-vector and $y$ an $l$-vector. Then $x \vee y$ is a vector of grade $n - ((n-k) + (n-l)) = k + l - n$ (and zero if $k + l < n$). Given that the join is bilinear, the inputs transform as $(-1)^{k+l}$ under the point reflection, while the transformed output gets a sign $(-1)^{k+l-n}$. Thus for odd $n$, the join is not $\mathrm{Pin}(n, 0, 0)$ equivariant.* ∎

To address this, given a pseudoscalar $z = \lambda\mathcal{I}$, we can create an equivariant Euclidean join via:

$$\mathrm{EquiJoin}(x, y, z = \lambda\mathcal{I}) := \lambda(x \vee y) = \lambda(x^* \wedge y^*)^{-*}. \quad (5)$$

**Proposition 6.** *In Euclidean algebra $\mathbb{G}_{n,0,0}$, the equivariant join $\mathrm{EquiJoin}$ is $\mathrm{Pin}(n, 0, 0)$ equivariant.*

*Proof: The $\mathrm{EquiJoin}$ is a multilinear operation, so for $k$-vector $x$ and $l$-vector $y$, under a point reflection, the input gets a sign $(-1)^{k+l+n}$ while the output is still a $k + l - n$ vector and gets sign $(-1)^{k+l-n}$. These signs differ by even $(-1)^{2n} = 1$ and thus $\mathrm{EquiJoin}$ is $\mathrm{Pin}(n, 0, 1)$-equivariant.* ∎

We prove two equalities of the Euclidean join which we use later.

**Lemma 6.** *In the algebra $\mathbb{G}_{n,0,0}$, let $v$ be a vector and $x, y$ be multivectors. Then*

$$v \rfloor (x \vee y) = (v \rfloor x) \vee y \quad (6)$$

*and*

$$x \vee (v \rfloor y) = -(-1)^n \widehat{v \rfloor x} \vee y. \quad (7)$$

*Proof: For the first statement, let $a$ be a $k$-vector and $b$ an $l$-vector. Then note the following two identities:*

$$a \vee b = \langle a^* b\tilde{\mathcal{I}}\rangle_{2n-k-l}\mathcal{I} = \langle a^* b\rangle_{n-(2n-k-l)}\tilde{\mathcal{I}}\mathcal{I} = \langle a^* b\rangle_{k+l-n}$$
$$= a^* \rfloor b,$$
$$(v \rfloor a)^* = \langle va\rangle_{k-1}\tilde{\mathcal{I}} = \langle va\tilde{\mathcal{I}}\rangle_{n-k+1} = \langle va^*\rangle_{n-k+1}$$
$$= v \rfloor (a^*).$$

*Combining these and the associativity of $\rfloor$ gives:*

$$(v \rfloor a) \vee b = (v \rfloor a)^* \rfloor b = v \rfloor (a^*) \rfloor b = v \rfloor (a \vee b)$$

*For the second statement, swapping $k$-vector $a$ and $l$-vector $b$ incurs $a \vee b = (a^* \wedge b^*)^{-*} = (-1)^{(n-k)(n-l)}(b^* \wedge a^*)^{-*} = (-1)^{(n-k)(n-l)}(b \vee a)$. Then we get:*

$$a \vee (v \rfloor b) = (-1)^{(n-k)(n-l-1)}(v \rfloor b) \vee a$$
$$= (-1)^{(n-k)(n-l-1)}v \rfloor (b \vee a)$$
$$= (-1)^{(n-k)(n-l-1)+(n-k)(n-l)}v \rfloor (a \vee b)$$
$$= (-1)^{(n-k)(n-l-1)+(n-k)(n-l)}(v \rfloor a) \vee b$$
$$= (-1)^{(n-k)(2n-2l-1)}(v \rfloor a) \vee b$$
$$= (-1)^{k-n}(v \rfloor a) \vee b$$
$$= -(-1)^{k-1-n}(v \rfloor a) \vee b$$
$$= -(-1)^n \widehat{(v \rfloor a)} \vee b.$$

*This generalizes to multivectors $x, y$ by linearity.* ∎

*b) Dual and join in projective algebra:* For the projective algebra $\mathbb{G}_{n,0,1}$ with its degenerate inner product, the dual definition of Eq. 4 unfortunately does not yield a bijective dual. For example, $e_0\widetilde{e_{012...n}} = 0$. For a bijective dual that yields the complementary indices on basis elements, a different definition is needed. Following Dorst [9], we use the right complement. This involves choosing an orthogonal basis and then for a basis $k$-vector $x$ to define the dual $x^*$ to be the basis $n + 1 - k$-vector such that $x \wedge x^* = \mathcal{I}$, for pseudoscalar $\mathcal{I} = e_{012...n}$. For example, this gives dual $e_{01}^* = e_{23}$, so that $e_{01} \wedge e_{23} = e_{0123}$.

This dual is still easy to compute numerically, but it can no longer be constructed solely from operations available to us in the geometric algebra. This makes it more difficult to reason about equivariance.

**Proposition 7.** *In the algebra $\mathbb{G}_{n,0,1}$, the join $a \vee b = (a^* \wedge b^*)^{-*}$ is equivariant to $\mathrm{Spin}(n, 0, 1)$.*

*Proof: Even though the dual is not a $\mathbb{G}_{n,0,1}$ operation, we can express the join in the algebra as follows. We decompose a $k$-vector $x$ as $x = t_x + e_0 p_x$ into a Euclidean $k$-vector $t_x$ and a Euclidean $(k-1)$-vector $p_x$. Then Dorst [9, Eq (35)] computes the following expression*

$$(t_x + e_0 p_x) \vee (t_y + e_0 p_y)$$
$$= ((t_x + e_0 p_x)^* \wedge (t_y + e_0 p_y)^*)^{-*}$$
$$= t_x \vee_{\mathrm{Euc}} p_y + (-1)^n \widehat{p_x} \vee_{\mathrm{Euc}} t_y + e_0(p_x \vee_{\mathrm{Euc}} p_y), \quad (8)$$

*where the Euclidean join of vectors $a, b$ in the projective algebra is defined to equal the join of the corresponding vectors in the Euclidean algebra:*

$$a \vee_{\mathrm{Euc}} b := ((a\widetilde{e_{12...n}}) \wedge (b\widetilde{e_{12...n}}))e_{12...n}$$

*The operation $a \vee_{\mathrm{Euc}} b$ is $\mathrm{Spin}(n, 0, 0)$ equivariant, as discussed in Lemma 5. For any rotation $r \in \mathrm{Spin}(n, 0, 1)$ (which is Euclidean), we thus have $r[a \vee_{\mathrm{Euc}} b] = r[a] \vee_{\mathrm{Euc}} r[b]$. This makes the PGA dual in Eq. (8) equivariant to the rotational subgroup $\mathrm{Spin}(n, 0, 0) \subset \mathrm{Spin}(n, 0, 1)$.*

*We also need to show equivariance to translations. Let $v$ be a Euclidean vector and $\tau = 1 - e_0 v/2$ a translation. Translations act by shifting with $e_0$ times a contraction: $\tau[x] = x - e_0(v \rfloor x)$. This acts on the decomposed $x$ in the following way: $\tau[t_x + e_0 p_x] = \tau[t_x] + e_0 p_x = t_x + e_0(p_x - v \rfloor t_x)$.*

*We thus get:*

$$\tau[x] \vee \tau[y]$$
$$= (\tau[t_x] + e_0 p_x) \vee (\tau[t_y] + e_0 p_y)$$
$$= (t_x + e_0(p_x - v \rfloor t)) \vee (t_y + e_0(p_y - v \rfloor t_y))$$
$$= x \vee y - t_x \vee_{\text{Euc}} (v \rfloor t_y) - (-1)^n \widehat{v \rfloor t_x} \vee_{\text{Euc}} t_y$$
$$\quad - e_0 \left( p_x \vee_{\text{Euc}} (v \rfloor t_y) + (v \rfloor t_x) \vee_{\text{Euc}} p_y \right)$$
$$\qquad\qquad\qquad\qquad\qquad \textit{(used (8) \& linearity)}$$
$$= x \vee y - e_0 \left( p_x \vee_{\text{Euc}} (v \rfloor t_y) + (v \rfloor t_x) \vee_{\text{Euc}} p_y \right)$$
$$\qquad\qquad\qquad\qquad\qquad\qquad\qquad \textit{(used (7))}$$
$$= x \vee y - e_0 \left( -(-1)^n \widehat{v \rfloor p_x} \vee_{\text{Euc}} t_y + (v \rfloor t_x) \vee_{\text{Euc}} p_y \right)$$
$$\qquad\qquad\qquad\qquad\qquad\qquad\qquad \textit{(used (7))}$$
$$= x \vee y - e_0 \left( (-1)^n (v \rfloor \widehat{p_x}) \vee_{\text{Euc}} t_y + (v \rfloor t_x) \vee_{\text{Euc}} p_y \right)$$
$$= x \vee y - e_0 (v \rfloor \left\{ (-1)^n \widehat{p_x} \vee_{\text{Euc}} t_y + t_x \vee_{\text{Euc}} p_y \right\})$$
$$\qquad\qquad\qquad\qquad\qquad\qquad\qquad \textit{(used (6))}$$
$$= \tau[x \vee y].$$

*The join is thus equivariant[3] to translations and rotations and is therefore $\mathrm{Spin}(n,0,1)$ equivariant.* ∎

Similar to the Euclidean case, we obtain full $\mathrm{Pin}(n,0,1)$ equivariance via multiplication with a pseudoscalar. We thus also use the EquiJoin from Eq. (5) in the projective case.

*3) Expressivity:* As also noted in Ref. [9], in the projective algebra, the geometric product itself is unable to compute many quantities. It is thus insufficient to build expressive networks. This follows from the fact that the geometric product preserves norms.

**Lemma 7.** *For the algebra $\mathbb{G}_{n,0,r}$, for multivectors $x, y$, we have $\|xy\| = \|x\| \|y\|$.*

*Proof:* $\|xy\|^2 = xy\widetilde{xy} = xy\tilde{y}\tilde{x} = x\|y\|^2\tilde{x} = x\tilde{x}\|y\|^2 = \|x\|^2\|y\|^2$. ∎

Hence, any null vector in the algebra can never be mapped to a non-null vector, including scalars. The projective algebra can have substantial information encoded as null vector, such as the position of points. This information can never influence scalars or null vectors. For example, there is no way to compute the distance (a scalar) between points just using the projective algebra. In the GATr architecture, the input to the MLPs that operate on the scalars, or the attention weights, thus could not be affected by the null information, had we only used the geometric product on multivectors.

To address this limitation, we use besides the geometric product also the join. The join is able to compute such quantities. For example, given the Euclidean vector $e_{12\ldots n}$,

---

[3]The authors agree with the reader that there must be an easier way to prove this.

we can map a null vector $x = e_{012\ldots k}$ to a non-null vector $x \vee e_{12\ldots n} \propto e_{12\ldots k}$.

### B. Architecture

In this section, we provide some details on the GATr architecture that did not fit into the main paper.

*a) Equivariant join:* One of the primitives in GATr is the equivariant join $\mathrm{EquiJoin}(x, y; z)$, which we define in Eq. (5). For $x$ and $y$, we use hidden states of the neural network after the previous layer. The nature of $z$ is different: it is a reference multivector and only necessary to ensure that the function correctly changes sign under mirrorings of the inputs. We find it beneficial to choose this reference multivector $z$ based on the input data rather than the hidden representations, and choose it as the mean of all inputs to the network.

*b) Auxiliary scalars:* In addition to multivector representations, GATr supports auxiliary scalar representations, for instance to describe non-geometric side information such as positional encodings or diffusion time embeddings. In most layers, these scalar variables are processed like in a standard transformer, with two exceptions. In linear layers, we allow for the scalar components of multivectors and the auxiliary scalars to freely mix. In the attention operation, we compute attention weights as

$$\mathrm{Softmax}_i \left( \frac{\sum_c \langle q_{i'c}^{MV}, k_{ic}^{MV} \rangle + \sum_c q_{i'c}^s k_{ic}^s}{\sqrt{8n_{MV} + n_s}} \right), \qquad (9)$$

where $q^{MV}$ and $k^{MV}$ are query and key multivector representations, $q^s$ and $k^s$ are query and key scalar representations, $n_{MV}$ is the number of multivector channels, and $n_s$ is the number of scalar channels.

### C. $n$-body dynamics prediction

*a) Dataset:* We first demonstrate GATr on a $n$-body dynamics prediction problem. Given the masses, initial positions, and velocities of a star and a few planets, the goal is to predict the final position after the system has evolved under Newtonian gravity for some time.

To be more precise, we generate data (for $n$ objects) as follows:

1) The masses of $n$ objects are sampled from log-uniform distributions. For one object (the star), we use $m_0 \in [1, 10]$; for the remaining objects (the planets), we use $m_i \in [0.01, 0.1]$. (Following common practice in theoretical physics, we use dimensionless quantities such that the gravitational constant is 1.)
2) The initial positions of all bodies are sampled. We first use a heliocentric reference frame. Here the initial positions of all bodies are sampled. The star is set to the origin, while the planets are sampled uniformly on a plane within a distance $r_i \in [0.1, 1.0]$ from the star.
3) The initial velocities are sampled. In the heliocentric reference frame, the star is at rest. The planet velocities are determined by computing the velocity of a stable circular orbit corresponding to the initial positions and masses,

| Parameter | GATr | Transformer | MLP | SEGNN |
|---|---|---|---|---|
| Layers | 10 blocks | 10 blocks | 10 layers | n/a |
| Channels | 16 multivectors + 128 scalars | 384 | 384 | n/a |
| Attention heads | 8 | 8 | n/a | n/a |
| Parameters [$10^6$] | 1.9 | 11.8 | 1.3 | 0.1 |

TABLE III: Hyperparameters used in the $n$-body experiments.

and then adding isotropic Gaussian noise (with standard deviation 0.01) to it.

4) We transform the positions and velocities from the heliocentric reference frame to a global reference frame by applying a random translation and rotation to it. The translation is sampled from a multivariate Gaussian with standard deviation 20 and zero mean (except for the domain generalization evaluation set, where we use a mean of $(200, 0, 0)^T$). The rotation is sampled from the Haar measure on $SO(3)$. In addition, we apply a random permutation of the bodies.

5) We compute the final state of the system by evolving it under Newton's equations of motion, using Euler's method and 100 time steps with a time interval of $10^{-4}$ each.

6) Finally, samples in which any bodies have traveled more than a distance of 2 (the diamater of the solar system) are rejected. (Otherwise, rare gravitational slingshot effects dominate the regression loss and all methods become unreliable.)

We generate training datasets with $n = 4$ and between 100 and $10^5$ samples; a validation dataset with $n = 4$ and 5000 samples; a regular evaluation set with $n = 4$ and 5000 samples; a number-generalization evaluation set with $n = 6$ and 5000 samples; and a E(3) generalization set with $n = 4$, an additional translation (see step 4 above), and 5000 samples.

All models are tasked with predicting the final object positions given the initial positions, initial velocities, and masses.

*b) Models:* Our GATr model is explained in III. We embed object masses as scalars, positions as trivectors, and velocities (like translation vectors) as bivectors.

GATr is compared to three baselines: the equivariant SEGNN [4], a vanilla transformer, and an MLP. For SEGNN, we use the code published by Brandstetter et al. [4] and the hyperparameters that publication uses for $n$-body experiments. We vary the number of nearest neighbours between 3 and the number of objects in the scene (corresponding a fully connected graph) and show the best result. For the Transformer baseline, we follow a pre-layer normalization [1, 24] architecture with GELU activations [12] in the MLP block. For the MLP, we use GELU activations as well.

In Tbl. IV we show hyperparameter choices and parameter counts.

*c) Training:* All models are trained by minimizing a $L_2$ loss on the final position of all objects. We train for 50 000 steps with the Adam optimizer, using a batch size of 64 and exponentially decaying the learning rate from $3 \cdot 10^{-4}$ to $3 \cdot 10^{-6}$.

*d) Results:* In the left panel of Fig. 3 we show the prediction errors as a function of the number of training samples used. The MLP, which has the least strong inductive bias and treats the object positions and velocities as a single, structureless feature vector, performs poorly on this task. The transformer structures the data in terms of objects and is permutation-equivariant, but not aware of the geometry; it achieves a reasonable prediction accuracy when using the full training set. SEGNN, which is E(3)-equivariant, achieves a substantially better performance than the non-geometric baselines. Our GATr architecture outperforms all three, achieving an asymptotic performance on par with SEGNN while being clearly more sample-efficient. It is able to predict final positions with high accuracy even from just 100 training samples.

GATr also generalizes robustly out of domain, as we show in the middle and right panels of Fig. 1. When evaluated on a larger number of planets, the mean error becomes larger, as non-trivial gravitational interactions become more frequent, but GATr still outperforms the baselines. In particular, both GATr and the baseline transformer generalizes better than SEGNN, providing evidence that a softmax-based attention mechanisms is more robust to object number generalization than the message passing algorithm of SEGNN. Finally, the performance of the E(3)-equivariant GATr and SEGNN does not drop when evaluated on spatially translated data, while the non-equivariant baselines fail in this setting.

### D. Robotic planning through invariant diffusion

*a) Environment:* We use the block stacking environment from Janner et al. [15]. It consists of a Kuka robotic arm interacting with four blocks on a table, simulated with Py-Bullet [8]. The state consists of seven robotic joint angles as well as the positions and orientations of the four blocks. We consider the task of stacking four blocks on top of each other in any order. The reward is the stacking success probability and is normalized such that 0 means that no blocks are ever successfully stacked, while 100 denotes perfect block stacking.

*b) Dataset and data parameterization:* We train models on the offline trajectory dataset published by Janner et al. [15]. It consists of 11 000 expert demonstrations.

To describe the problem in terms of geometric quantities, we re-parameterize the environment state into the positions and orientations of the robotic endeffector as well as the four blocks. The orientations of all objects are given by two direction vectors. In addition, there are attachment variables that characterize whether the endeffector is in contact with either of the four

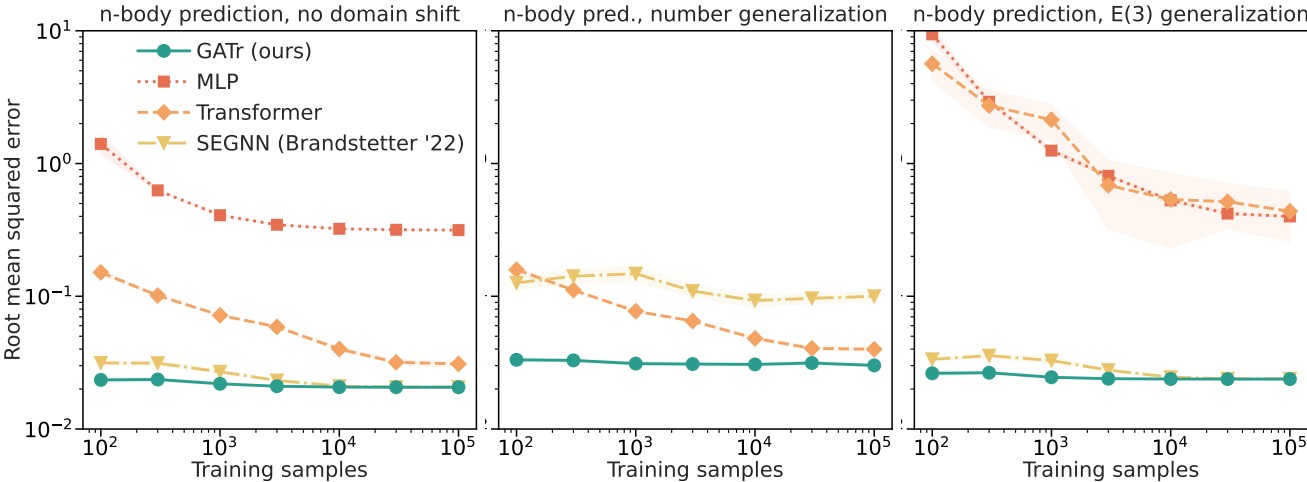

Fig. 3: Results on a synthetic $n$-body dynamics dataset. We show the error in predicting future positions of planets as a function of the training dataset size. Out of five independent training runs, the mean and standard error are shown. **Left**: Evaluating without distribution shift. GATr (──) is more sample efficient than SEGNN [4] (──·) and outperforms non-geometric baselines (━ ━, ·····). **Middle**: Evaluating on systems with more planets than trained on. Both GATr and the baseline transformer generalize well to different object counts. **Right**: Evaluating on translated data. Because GATr is E(3) equivariant, it generalizes under this domain shift.

| Parameter | GATr-Diffuser | Transformer-Diffuser | Diffuser |
|---|---|---|---|
| Transformer blocks | $\{10, \mathbf{20}, 30\}$ | $\{10, \mathbf{20}, 30\}$ | n/a |
| Channels | 16 multivectors + 128 scalars | $\{\mathbf{144}, 384\}$ | n/a |
| Attention heads | 8 | 8 | n/a |
| Parameters $[10^6]$ | $\{2.1, \mathbf{4.0}, 5.9\}$ | $\{1.8, \ldots, \mathbf{3.5}, \ldots, 35.7\}$ | 65.1 |

TABLE IV: Hyperparameters used in the robotic planning experiments. For GATr-Diffuser and the Transformer-Diffuser, we experimented with different depth and (for the Transformer-Diffuser) channel counts. For each model, we independently chose the best-performing setting, shown here in bold. The Diffuser model uses a substantially different architecture based on a U-net, we refer the reader to Janner et al. [15] for details.

blocks. In this parameterization, the environment state is 49-dimensional.

We train models in this geometric parameterization of the problem. To map back to the original parameterization in terms of joint angles, we use a simple inverse kinematics model that solves for the joint angles consistent with a given endeffector pose.

*c) Models:* Our GATr model is explained in Sec. III. We use the axial version, alternating between attending over time steps and over objects. We embed object positions as trivectors, object orientations as oriented planes, gripper attachment variables as scalars, and the diffusion time as scalars.

For the Transformer baseline, we follow a pre-layer normalization [1, 24] architecture with GELU activations [12] in the MLP block and rotary positional embeddings [22]. For the Diffuser baseline, we follow the architecture and hyperparameters described by Janner et al. [15].

For all models, we use the diffusion time embedding of Ref. [15]. In Tbl. IV we show hyperparameter choices and parameter counts.

All models are embedded in a diffusion pipeline as described by Ho et al. [14], using the hyperparameter choices of Ref. [15]. In particular, we use univariate Gaussian base densities and 1000 diffusion steps.

*d) Training:* We train all models by minimizing the simplified diffusion loss proposed by Ho et al. [14]. For our GATr models and the Diffuser baselines we use an $L_2$ loss and train for $200\,000$ steps with the Adam optimizer, exponentially decaying the learning rate from $3 \cdot 10^{-4}$ to $3 \cdot 10^{-6}$. This setup did not work well for the Diffuser model, where (following Janner et al. [15]) we use a $L_1$ loss and a low constant learning rate instead.

*e) Evaluation:* All models are evaluated by rolling out at least 200 episodes in a block stacking environment and reporting the mean task and the standard error. We use the planning algorithm and parameter choices of Janner et al. [15] (we do not optimize these, as our focus in this work is on architectural improvements). It consists of sampling trajectories of length 128 from the model, conditional on the current state; then executing these in the environment using PyBullet's PID controller. Each rollout consists of three such phases.