# OpenReview forum: "Geometric Algebra Transformers"
_roboticsfoundation.org/RSS/2023/Workshop/Symmetry — RSS 2023 Workshop Symmetry_

### Official Review · Reviewer_RF6L · 2023-06-16

**Rating:** 7
**Confidence:** 3

**Review:**

The paper proposes an E(3) equivariant transformer architecture, GATr, where the representations are expressed in the projective geometric algebra.  The paper describes a framework for embedding inputs, such as scalars, vectors and rotations, in the projective geometric algebra, and introduces equivariant linear and non linear mappings that act on embeddings of the geometric algebra.  The proposed model outperforms baselines in a robotic planning task and a dynamics prediction task.  I believe the idea is interesting and is a good fit for this workshop.

Strengths
- The paper is well written and introduces the geometric algebra in an easy-to-understand way.
- The method outperforms baselines in both experiments, with significantly better performance in the low-data regime.  The authors are careful to make the comparison fair by re-running baselines with the same data parametrization.
- The idea to use the geometric algebra to design an E(3) equivariant network is novel and interesting.  While several variations of equivariant attention and equivariant layer norms have been proposed, the geometric bilinear layer is a non-trivial extension.  An ablation study to highlight the importance of this operation would be a good addition.

Weaknesses
- The paper does not discuss any limitations of using the projective geometric algebra.  For instance, are there any 3D inputs that could not be easily mapped to the geometric algebra?  Could it scale to more complex inputs like point clouds?
- The experiment section could be strengthened by including comparisons to other equivariant transformer methods, such as [1] or [2].  Currently, the paper does not describe how using geometric algebra differs from existing equivariant networks.

Questions
- Where does the 8 come from in the Attention equation (Eqn 3)?
- It is not fully clear why auxiliary scalars should be treated separately, since the geometric algebra includes scalars.


[1] Fuchs, Fabian, et al. "Se (3)-transformers: 3d roto-translation equivariant attention networks." Advances in Neural Information Processing Systems 33 (2020): 1970-1981.
[2] Assaad, Serge, et al. "VN-Transformer: Rotation-Equivariant Attention for Vector Neurons." arXiv preprint arXiv:2206.04176 (2022).

---

### Official Review · Reviewer_FkT1 · 2023-06-18
**Strong Accept**

**Rating:** 9
**Confidence:** 3

**Review:**

This paper introduced a new variant of the Transformer architecture imbued with a geometric inductive bias (i.e. E(3) equivariance) which yields performance and sample efficiency improvements. The geometric inductive bias is created by representing the input, output and hidden states of the proposed architecture in projective geometric algebra, implemented as a series of pre and post-processing steps as well as modified standard neural network components such as Linear, LayerNorm layers with equivariant properties. The authors also provided detailed proofs of the proposed properties of GATr. Performance improvement is demonstrated through a simulated robotic planning task and a n-body dynamics regression task.

Pro:
Novel architecture and theoretical result.
Large performance improvement on the low-data regime in both experiments.

Con:
Missing related works and limitations sections.
No experiment on real-world datasets (does the same inductive prior apply to real-world data?)

---

### Decision · Program_Chairs · 2023-06-24

**Decision:**

Accept

**Comment:**

Congratulations! We encourage the authors to revise the paper based on the reviewer's feedback.
Your paper will be presented as both a short presentation and a poster. Detailed instructions about the presentation format and camera-ready submission will be sent to you soon.